# Flow. A Socially Responsible 3D Printed One-Handed Recorder

**DOI:** 10.3390/ijerph182212200

**Published:** 2021-11-20

**Authors:** Javier Esclapés, Almudena Gómez, Ana Ibañez

**Affiliations:** 1ArtefactosLAB, Research Group “Design Engineering and Technological Development”, University of Alicante Polytechnic School, Crta. San Vicente s/n, San Vicente del Raspeig, 03690 Alicante, Spain; 2ArtefactosLAB, University of Alicante Polytechnic School, Crta. San Vicente s/n, San Vicente del Raspeig, 03690 Alicante, Spain; flow@artefactos.org; 3Research Group “Public Relations, Social Responsibility and Communication with the Specialized Population and Minorities”, University of Alicante, Crta. San Vicente s/n, San Vicente del Raspeig, 03690 Alicante, Spain; ana.ibanez@ua.es

**Keywords:** recorder, assistive technology (AT), music therapy, inclusive education, 3D printing, fair-cost, human-centered design, socially responsible, additive manufacturing, stereolithography

## Abstract

The recorder is one of the most common instruments used during primary school in the formal education system in the EU. However, there are a percentage of students with only one functional hand. The existing one-handed recorders available for them to be able to play and perform in the same way as their peers are expensive and difficult to use. This study’s purpose is to document the development of Flow—a low cost one-handed recorder as well as the user’s assessment of the psychosocial benefits of the recorder. The methods used for fabrication were 3D modelling and additive manufacturing (AM) technology or 3D printing using the technique of stereolithography, and for the assessment of the product, the Psychosocial Impact of Assistive Devices Scale (PIADS) questionnaire was distributed to 20 primary school users. The results show that the use of resins and Stereolithography is appropriate for wind instruments providing quality and strength at a fair price. Flow also proved to have a positive impact on the users and their inclusion in school. The main conclusions of this study underscore the adequacy of using AM for adaptations required for people with disabilities and the positive psychosocial benefits generated by the use of Flow in children.

## 1. Introduction

People with disabilities have to face extra costs resulting from such disability, such as costs associated with medical care or assistive devices. As a result of those higher costs, people with disabilities and their households are likely to be poorer than non-disabled people with similar incomes [1]. In Spain, the incomes of families with members who have a disability are 29.2% lower than the average families [2]. In addition, there are a lack of alternatives and the cost-quality ratio of adapted technologies is unsatisfactory [1].

The *United Nations Convention on the Rights of Persons with Disabilities* features countries promoting the necessary actions in order to ‘undertake or promote research and development, and promote the availability and use of new technologies, including health technologies, information and communications, mobility aids, technical devices and appropriate assistive technologies for people with disabilities, giving priority to those with an affordable price’ [3].

Therefore, it is necessary to increase the use and affordability of assistive devices and technology [1] and one of the options to lower costs and bringing appropriate assistive devices to users is additive manufacturing (AM) technology, more commonly known as three-dimensional (3D) printing. AM technology is a manufacturing process characterized for being accessible and for allowing low costs, rapid prototyping, and rapid manufacturing [4,5].

Within this context and after considering users’ needs and preferences, a group of families, engineers, therapists, students, and researchers, with the support of the research group DIDET (Engineering Design and Technological Development) from the University of Alicante (Spain), created a research project called Artefactos. The main aim of this project is the production of fair-cost assistive devices or adaptations for people with disabilities using additive manufacturing (AM) technologies [6]. Accordingly, this study presents the design, manufacturing, and user assessment of Flow the one-handed recorder, which is one of the projects developed by this group.

The need to design a recorder that can be played with only one hand was firstly suggested by families associated to the non-profit organizations HEMIweb, AFANIP and ADAYO in Spain. Commonalities with these families were that one of their members had only one functional hand and was enrolled in primary school where recorders are one of the most common instruments used for music education as part of the formal curriculum in the EU [7].

More than one billion people (one out of seven) in the world live with some form of disability and this represents 15% of the population [1]. Among this percentage, there are a large number of people with motor disabilities [3]. In Spain, there are 13,917 (6.3% of the total) students with motor disabilities, 87.7% of them are integrated in formal education, and 38.6% (5361) attend primary schools [8]. On the other hand, the number of people affected by infantile hemiparesis, which is one of the main reasons why only one hand is fully functional, is estimated at about 6750 children between 0 and 16 years of age in Spain [9]. Likewise, this number increases if other conditions that interfere in the functionality of both upper limbs or that generate difficulties with bimanual activities are included, such as hemiplegia, obstetric brachial plexus palsy, childhood acquired brain damage, cerebral palsy in general, amputations, and other congenital or cognitive conditions.

Thus, those families who are affected by the above-mentioned conditions need to acquire adapted recorders, which are difficult to use, expensive, and usually designed for professionals, if they want their children to be able to perform in the same way as their peers. The existence of devices adapted for kids with a single functional hand, especially for the learning of music, is limited in terms of diversity of models and not very accessible in terms of cost and places where they are available. Currently, the unimanual recorders available on the market are Yamaha YRS900 (EUR 400, only available in Japan), Dolmetsch Gold Soprano DGS (EUR 650) and Mollenhauer Pearwood (EUR 1199). Therefore, Artefactos was commissioned to develop a more accessible, suitable, high quality, and appropriate adapted recorder for primary school children.

“Some of the most successful one-handed instrument adaptations have been wind instruments. This has largely to do with the fact that both hands tend to play a similar role, i.e., pressing different combinations of buttons to open and close valves, while sound activation is achieved using the mouth. This playing method lends itself to a conceptually straightforward (albeit mechanically challenging) adaptation for one hand, whereby the mapping between valve closure and the buttons is reconfigured for a single hand.”[10]

Over the past two decades, researchers from various disciplines that intersect with music have engaged in hacking-like endeavours by adapting instruments for people identified as physically disabled. These include adaptations for the recorder [11,12,13,14], saxophone [15], and trumpet [16].

“Examples of successful one-handed wind instrument adaptations can be found on the One-Handed Musical Instrument (OHMI). Ref. [16] also summarises several earlier wind instrument adaptations for one hand or otherwise. For many wind instruments, successful one-handed adaptations have been achieved.”[10]

Regarding 3D printed one-handed recorders, the OHMI website includes a one-handed 3D-printed recorder and, in general, there are several attempts to apply 3D printing techniques for wind musical instruments [4,17,18], all of which have used polymer-based techniques. These studies prove the benefits of the wide capabilities of 3D printing in free-shape design [13,18] and it is a comfortable tool for geometrical development [17].

As mentioned before, Flow was manufactured by using AM, more specifically, according to the two processes, which are most widely used for its production. Those processes are, on the one hand, fused deposition modelling (FDM), which is used for prototypes, and is one of the most usual and common extrusion-based 3D printing methods. On the other hand, stereolithography (SLA), which fabricates 3D objects by selectively solidifying the resin through photopolymerisation and is commonly used for the production of final parts [19].

Another characteristic of AM is the “open source” nature of the designs that allow for versatility and customization, which are necessary for people with disabilities and custom-designed instruments, as well as for augmentations/modifications of existing instruments. The files used with AM can be adapted and easily customized, thus offering significant benefits over traditional, non-modifiable instruments [20,21]. Therefore, Flow files can be easily modified and adapted both in size and functions, as well as aesthetically, for example, for colours as well as the inclusion of decorative elements and the engraving of names according to specific user needs and preferences.

The low cost fabrication, easy manufacturing, and customization options of Flow makes it easy to acquire around the world, and of course, it is also suitable for adults and can be used in the field of rehabilitation and music-supported therapy when some type of assistive device is required to treat certain learning disorders.

Accessible instruments provide a means for people with disabilities to enjoy the wide range of health, social, and psychological benefits of music making. Accessible instruments also contribute to an inclusive education, such as education in the arts and cultural experiences, which are regarded [10,14,16,20,21] as an important part of contributing to social inclusion in society and towards the universal agenda of the Sustainable Development Goals (SDGs) to wipe out poverty through sustainable development by 2030 [22].

In February 2019, researchers from Artefactos LAB accepted the challenge to design this project, following human-centered design, using 3D printing and according to socially responsible ethics. Two years later, a valid proof of concept was reached and the assessment of the psychosocial impact of assistive technologies was performed using the PIADS [23].

The main aim of this research is to focus on people with disabilities through a product design that will respond to the final users’ needs. In order to achieve this goal, all the necessary agents and people who would benefit from it were actively involved in the development and design of Flow. The value and trust in Flow was achieved through responsible two-way communication with all individuals involved, and dues to active and careful listening to the needs of individuals. This contributed to provide the design team with constant and useful feedback. Flow is an example of the important role that social responsibility plays in scientific research in order to contribute to the progress of mankind. In addition, such responsibility is aligned with society values and leads researchers to attend to social demands and to offer alternative solutions aimed at facilitating progress, social change, and development in the environment [24].

## 2. Materials and Methods

### 2.1. Design Process

The methodology and development of Flow has followed the so-called ‘human-centered design’ [25,26,27]. Such methodology is a creative process, which firstly focuses the attention on the people for which the design is being conceived and, secondly, proposes new, innovative, and customized solutions. This methodology also promotes empathy with the potential users, facilitates the generation of ideas, involves the creation of prototypes, and motivates knowledge sharing.

The human-centered design consists of three stages:The inspirational stage: The designer immerses himself/herself in the user’s life while establishing a close relationship based on trust and reciprocity, which favours a better understanding of the user’s needs.In this stage, with the aim of gathering some information from potential users about some areas of interest, such as the teaching of music or the use of the recorder in the classroom, several interviews and surveys are conducted with children (and their families) as well as with some associations interested in the implementation of the project. Some of these associations are Child Hemiparesis (Hemiweb), Spanish Association of Professional Musicians (Ampemusicos) and the National Spanish Confederation of Associations of Musical Education (COAEM).The conceptualization stage: Design requirements are identified, and a prototype is developed. Section 2.2 and Section 3.1 shows the methodology and results of this stage, respectively.The implementation stage: The proposed solution is assessed and once it has been approved it is offered to the user [28,29,30,31]. The assessment of the psychosocial impact of assistive technologies is performed using the PIADS [23]. Section 2.3, Section 2.4 and Section 3.2 shows the methodology and results of this stage, respectively.

### 2.2. Product Development

#### 2.2.1. Software. 3D Modelling and Lamination for Additive Manufacturing

The parametric design software Autodesk Inventor [32] was used for the 3D modelling of the recorder. Such software allows the 3D parametric modelling of parts and a solid assembling in addition to the simulation and rendering of mechanical units. In addition, the software allows to export files in STL format, thus connecting with the additive manufacturing process. Free alternative software that could be used is FreeCAD [33].

The additive manufacturing process requires a lamination software, which allows the positioning and orientation of the 3D model in the machine, the optimization of the manufacturing parameters (the filling pattern, the material, the supports, etc.) as well as the possibility to export the file to the corresponding manufacturing machine. In this regard, two technologies have been used: fused deposition modelling (FDM), which uses the free software Cura of Ultimaker [34]; and stereolithography technology (SLA), which uses the software PreForm from Formlabs [35].

#### 2.2.2. Hardware. Additive Manufacturing and Materials

The low price of both the materials and the equipment required would explain the increasing widespread use of additive manufacturing as an alternative to the usual process, and the fast integration of this process in some production sectors. This fact, along with the freedom of design and the incredible versatility offered by three-dimensional modelling systems have allowed technical or industrial problems as well as health problems related to people’s quality of life to be solved. Such problems could not be solved without increasing the costs (due to the low number of manufactured units required), thus making the manufacturing of such devices not viable [36].

The use of these additive techniques allows for a better approach to users’ personal and intrinsic needs in the sector of assistive technologies, which go beyond the clinical or therapeutic needs. Such technologies offer the user with functional diversity additional aspects that will improve his/her quality of life while taking into consideration the empathy and the communication with the user in the processes of production and use of assistive devices (through the use of accessible design and technology) [37]. The technologies used in the manufacturing of the adapted recorder follow.

Fused deposition modelling (FDM) is a process of additive manufacturing used for the modelling of prototypes and for small-scale production. FDM allows the use of production thermoplastics for the manufacturing of resistant, durable, and dimensionally stable parts while offering quality and a personalized final product at a fair price [37]. For the manufacturing of Flow prototypes, the type Ender 3 from Creality (Shenzhen, China) was used with a standard nozzle of 0.4 mm for the extrusion of rigid Polylactic Acid (PLA) filaments with different properties (Figure 1). The same machine was used for the extrusion of flexible parts, although it was adapted with a special system of direct extrusion for flexible filaments of thermoplastic elastomers (TPE), more specifically, the type V3 from Recreus (Alicante, Spain).

Stereolithography (SLA) has been used for printing as this technology provides personalised design instruments (shape and texture) and superior mechanical properties (lightweight and strength), which, in turn, have acoustic properties suitable for school use [18].

In the SLA fabrication process, ‘the approach of stereolithography is carried out through a localised photopolymerisation process, which is triggered by ultraviolet (UV) radiation and takes place within a bath containing liquid monomers, oligomers, and photoinitiators. The base of stereolithography is the curing reaction of resins, which is an exothermic polymerization process characterized by chemical cross-linking reactions. The reaction is initiated by supplying the energy of UV light, and there are two transitions during the curing reaction process: gelation and vitrification. Gelation is a liquid-to-rubber transition that produces a dramatic increase in viscosity. During this transition, both gel phase and sol phase coexist in the system. Vitrification is a gradual, thermo-reversible process that leads to the transition from liquid or rubber resin to glassy solid resin [19].

The prototypes and the final product were manufactured through stereolithography (SLA), which is one of the additive manufacturing (AM) technologies. The machine Formlabs 3 (Somerville, MA, USA) was used with the Form Wash machine to clean parts (Figure 2).

The chosen material was Tough 2000 Resin. This is the strongest and most rigid material of the functional family of Engineering Resins [38], and it has been used for resistant pieces that should not bend easily. The benefits of this technology and material are: high quality appearance, strong and rigid prototypes, templates and accessories that require minimal deflection and that simulate the strength and rigidity of acrylonitrile–butadiene–styrene (ABS) polymer.

#### 2.2.3. Acoustic/Musical Properties

As has been mentioned previously, the main purpose of this social device is to improve the psychosocial aspects of children at school (without any professional purpose). For this reason, we measured and analyzed the musical and sound basic properties of Flow, and the frequency of music notes using the gStrings application. It was also examined and compared with a Yamaha one-hand recorder (Yamaha YRS-900) [18].

### 2.3. Instruments and Variables

The instrument used to evaluate Flow was the Psychosocial Impact of Assistive Device Scale PIADS [23] and a cross-sectional design conducted with a sample of 20 participants from primary school (from 7 to 11 years old with disability in one hand). PIADS consists of 26 items, each one of them describing a possible effect of using the assistive device. The user is asked to rate how he or she feels about each item on a scale from −3 (indicating the most negative impact) to +3 (indicating the most positive impact). For example, the user is asked to rate how the assistive device affects his or her ‘sense of control’ or ‘willingness to take chances’. Scoring on the PIADS yields three subscale scores that were derived through factor analysis and a total score. The competence subscale consists of 12 items, the adaptability scale consists of 6 items, and the self-esteem scale consists of 8 items. Each of the subscales, as well as the total score, is computed by obtaining the mean of all items within that subscale or full scale, so that each subscale score ranges from −3 to +3. Internal consistency is highly reliable, as demonstrated by Cronbach alpha of 0.95 for the total score and 0.92, 0.88, and 0.87 for the competence, adaptability, and self-esteem subscales, respectively [39,40]

In addition to the psychosocial variables, other aspects directly related to features of how the product works such as innovation, usability, quality of sound, reliability, ergonomics, grip, and cost have also been analysed. Such analysis has been carried out through surveys, which have included different questions that the user or the expert could answer using a Likert scale (from 1 to 5) [41].

### 2.4. Sample

The participants in this survey are members of the non-profit organizations (Hemiweb, AFANIP and ADAYO), and they have been diagnosed with hemiplegia, agenesis of the arm, and obstetric brachial plexus injury (OBPI) (see Table 1). This information was collected from the observations made by the users in the PIADS and in the questionnaires about the features of the recorder. The inclusion criteria were school-age individuals with only one damaged hand or absent hand (either left or right).

### 2.5. Ethical Consideration

This project is fully adapted to the current strategies of socially responsible research, which seeks the empowerment of excluded groups through the design and development of technical assistance, at both international and national levels. In this sense, it is important to highlight the recent approval of the Bologna Declaration promoted by the Association for the Advancement of Assistive Technology in Europe (AAATE) [42], which includes, among others, the following proposals:To promote in all relevant disciplines socially responsive and responsible research, the investigation of existing barriers preventing full inclusion of all in society as well as developing strategies and solutions—many of which may be technology-related—to enable participation.To assure that technological innovation takes into account the greatest possible number of potential beneficiaries following a universal design approach, and make sure that it does not contribute to further exclusion by widening the gap between the haves and have-nots.To foster assistive technology provision systems that are person-centered, independent from commercial interests, and able to provide, in a timely and affordable manner, personalised forward-looking solutions that are suitable for the environment of use and based on the abilities, preferences and expectations of the final user.To pursue and assure the quality of assistive technology solutions for the equitable and global provision of assistive technology systems.To promote positive images, designs and initiatives, which counteract the stigma that sometimes is associated with impairment and the use of assistive technology.

In conclusion, Flow is a unique opportunity to advance scientifically and socially in the creation of an adapted recorder that would have a substantial international and social impact on those individuals with dysfunctional hands and without the possibility to play recorder at school.

On the other hand, the development of this product is aligned with ODS number 10 regarding ‘reducing inequality’ among vulnerable groups, especially those with disabilities, due to their risk of exclusion [22].

## 3. Results

### 3.1. Design and Development Results

Following initial interviews (inspirational stage) and taking into account the specific benefits of music for inclusion at schools [20,21,43,44] as well as the possibilities to produce customised instruments through additive manufacturing at a fair price [45,46], it was decided to start with the design and development of an adapted recorder to use at schools.

After several interactions carried out during the process of design and development, a functional and valid concept has been proposed. The Flow concept consists of fewer parts, shows higher quality, and is planned to be manufactured by SLA. Table 2 displays the features, relevant specifications, benefits, and limitations of the recorder concept developed and proposed in this study, for the adapted recorder in comparison to the already existing adapted recorder by Yamaha (type YRS900).

#### 3.1.1. Design Results

Flow responds to design optimisation, reduction of the parts, and prioritisation of the quality of the instrument. This has resulted in a type of adapted descant recorder, which can be played with one hand and only by using five fingers of the same hand (either left or right)—four fingers to open and close the holes, and the fifth finger to hold the recorder.

Figure 3 shows some of the conceptualizations for this prototype whereby only three keys are used to open the three holes at the top. It also displays a visual study on the ergonomics of the instrument’s grip.

This concept of the adapted recorder has been conceived to be played with one hand due to a combination of keys and a specific setting when pressing those keys along with the closing or non-closing of the rest of the holes. As displayed in Figure 4, this prototype consists of a body: (1) with two ends: one at the top to assemble the mouthpiece and one at the bottom of the recorder (5) where the trim (4) and the non-slip part are inserted (6). Both ends have an adjustable rubber band attached (2) and (3). Additionally, the keys (9), magnet cases (7), grip ring (13) and rubber rings (11) are assembled on the body of the recorder. The keys and the grip ring stay attached and connected to the body through a metal rod (12) and (14). The inner design of the parallelepiped protruding parts located at the body of the recorder allows the rod to stay attached to the whole instrument. The rubber bands attached to the body of the recorder and to the keys, on the one hand, and the magnets (8) and (10), which are inserted in both the cases and the keys, on the other hand, allow the whole system of the recorder to display a starting position where the holes are closed. Such a position, in which three of the holes are already closed, allows for playing the musical note G without using any fingers. By blowing the mouthpiece and by pressing with only one hand the different combinations of keys, the following musical notes can be played: C, D, E, F, G, A, B, C′, D′, E′, F sharp, and B flat.

The design of the grip ring is based not only on the user’s individual features but also according to the needs resulting from the action-interaction with the instrument. The grip ring is assembled to the body of the recorder through a rod, which allows the user to hold the instrument with just the thumb, thus still having four fingers to play the musical notes. The grip ring includes a mounting device allowing the rod to partially rotate on its own axis, thus providing more stability when holding the instrument as well as more movement and safety when using it.

As mentioned before, the adapted recorder has been manufactured by SLA using rigid resin (Tough from Formlabs (Somerville, MA, USA)). Therefore, it is possible to customise it and to manufacture it in the industrial sector according to user’s needs. The grip ring is preferably manufactured with flexible filament, since the qualities and finishing of flexible filament will provide the recorder with the flexibility, smoothness, ergonomics, and durability, which are required for its effective functioning.

From a technical point of view, this system also improves the features and functions of the already existing recorders in the market by reducing the weight, making the learning easier or providing an easy and faster way of acquiring spare parts (Figure 5 and Figure 6).

The production of this Flow concept has an estimated cost of 140 euros. The material, manufacturing, and assembly processes increase the price of this prototype. Even so, in order for all families in need to have a recorder, Artefactos (Hemiweb, or another association) will offer the recorder on free loan.

#### 3.1.2. Acoustic Results

The musical and sound properties of Flow were examined and compared with a Yamaha one-hand recorder (Yamaha YRS-900). The testing of musical notes was performed using gStrings application.

According to gStrings application tests, the Flow concept has all the notes within reach and in the appropriate positions (see Table 3). This result allows for concluding that it is possible to proceed to test Flow by experts in musical education.

#### 3.1.3. Views from Experts in Musical Education at Schools

Table 4 shows the results of the assessment conducted by a group of experts in music at schools. The group consisted of 15 primary school music teachers.

This survey was conducted in October 2020 and each music teacher participating in the survey was provided with the two types of adapted recorders, namely, Yamaha YSR-900, and Flow concept. First, the music teachers were given several days to trial the two types of adapted recorders. Then, the questionnaires were provided [41] and were completed by the end of 2020. Figure 7 shows through a radar chart the results of this survey.

According to musicians’ views, the Flow concept was preferred mainly because it demonstrated an innovative design, was easy to use, allowed to grip with just one hand, deemed comfortable, and was reliable. Moreover, most of the participants characterised the musical sound of Flow as ‘very good’.

### 3.2. User’s Perspective

Initially, the users (20 in all) received and tested the Flow concept in January 2021 at home (Figure 8). Three months later, the questionnaire to assess the main features of the recorder with Likert scale [41] and PIADS questionnaire [39] were sent to the participants’ families. Finally, answers were collected from May to June 2021.

Analysis of results (Figure 9) has concluded that all aspects related to design and innovation scored high, while usability and comfort obtained an average score, and the cost of the instrument was rated low (44%).

For the processing of the PIADS results, demographic, gender, and clinical variables were excluded because of the sample size. Descriptive statistics for the PIADS and results are displayed in Table 5.

Subscales of PIADS (competency, adaptability, and self-esteem) scored high, meaning that the impact of Flow recorder on the lives of the kids participating in this study could be regarded as very positive.

## 4. Discussion

According to the tests that have been carried out and the results of the questionnaires on assessment of the proposed designs, it can be concluded that the design concept of Flow is the final and most appropriate for this project. The main goals that have been achieved through this final design, and which have been validated by the assessment discussed in Section 3.1 and Section 3.2 of this article are as follows:Adaptability: Both users and music teachers show a medium-high degree of satisfaction with features related to the adaptability of the product (usability, comfort, and grip; Figure 7 and Figure 9). This is most probably due to the methodology used, which is user-centered [29,30], thus allowing to achieve the biggest challenge of this project, namely, to design a recorder that can be played with one hand.User-friendly: Both users and music teachers show a medium-high degree of satisfaction in terms of how user friendly the recorder is (reliability, grip, usability, and comfort; Figure 7 and Figure 9). This user friendliness has been possible due to the simple design of the key system [47,48], and the exclusion of the thumbhole, which is not necessary at the school level for which the recorder is designed.Good acoustics and tuning: the acoustic tests (Table 3) and the assessment provided by music teachers (Figure 7) show the good quality of the recorder’s acoustics. This might be linked to the specific benefits of resin, and its special acoustic features, which have been discussed in other articles [18].Durable: It was important to propose a design that will consider and respond to children’s activity. Therefore, it was decided to use an impact-resistant material. As shown in the assessment, both users and music teachers show a high degree of satisfaction in this regard (reliability and comfort; Figure 7 and Figure 9).Lightweight: The simple design and the properties of the material (resin Tough 2000) have allowed the design of a lightweight and slim device. This feature prevents overload while using or carrying the recorder.Fair cost: Additive manufacturing allows for the production of a tailored-made and customized device, which can be produced in a short period [36,47]. In addition to this, the device does not require a great initial investment in technology, manufacturing, or storage. Accordingly, the user is offered a high quality instrument for an estimated price of EUR 140 (as opposed to the initial EUR 400). Such price would cover the production cost (without being overly costly).Psychosocial aspects: As analysis of results show in Table 5, learning and playing with this recorder have clear psychological benefits for one-handed children. Users and families point out the clear benefits of playing this recorder at a social level, as well as with classmates and friends, thus promoting one of the most important goals of this research, namely, inclusion at schools [20,21,49,50].

### Further Research

The feature that scored the lowest in the assessment of the instrument was the cost of the device (Figure 7 and Figure 8); therefore, it is a matter of concern for families, experts, and developers. A short-term solution to this issue is an agreement reached with associations of users (such as Hemiweb) so that Flow is included in the adapted-recorder loan scheme, for which such associations have been offering for some time now. However, as a long-term solution, some arrangements have been initiated for the manufacturing and assembly of the recorder at industrial scale. This is thought to remarkably lower the cost of the instrument.

As far as the views from users and families on the positive impact of adapted instruments at schools is concerned, it would be relevant to carry out more comprehensive psychosocial studies in order to determine the impact of using the adapted recorder with the entire class. This would imply including all students in the research in order to see whether inclusion at schools promotes positive emotional values in the group, and even whether it improves school performance of the whole group, as stated in previous studies.

Finally, it is important to highlight that initial tests to adapt the recorder to adults have been carried out. In doing so, people with functional diversity in the upper limb who wish to learn how to play this instrument, will be able to do it. Such an adaptation for adults would only imply to enlarge the inner side of the thumb ring of the adapted recorder so that it matches the shape and size of an adult hand. In this case, the instrument should be validated with a studio recording test.

## 5. Conclusions

The aim of this paper is to propose the design and development of an adapted instrument Flow, which has been proven to be beneficial for the psychosocial wellbeing of children with functional diversity in the upper limb. This design also shows the possibilities of new technologies of additive manufacturing (SLA or FDM) to produce advanced assistive devices, which are reliable, show high performance, and are produced at a low cost. All these features are key for the inclusion of people with functional diversity (Figure 10) and for the improvement of their quality of life.

This research contributes to the use of the above-mentioned technologies in the design and development of further adapted instruments or new high performance musical instruments. Such contribution aligns with the idea of responsible scientific research promoted by universities and the impact of such research in solving specific issues.

In this sense, this research proves the effectiveness of applying the human-centered methodology, which is the approach taken by the Laboratory Artefactos at the University of Alicante for the design and manufacturing of assistive devices, which respond to society’s needs through responsible research.

Such commitment with responsible research is based on social relevance and social permeability, and shows the need for open communication with social agents and all people involved. It is also important that both those social agents and the people involved are active participants in solving the issues that might have been identified by researchers, thus contributing to the optimal development of the product. Additionally, user’s involvement in the development of the product empowers the group for which the product Flow is being designed, improves their self-esteem, and favours both their social inclusion and inclusion at schools.

## 6. Patents

This research has led to the registration of the following two utility models:Esclapés Jover, J., and Gomez Hernansanz, A. (2021). Patente n° U202131434. España.Esclapés Jover, J., Gómez Hernansanz, A., and Cabrera Madrigal, L. (2019). Patente n° PCT201932022. España.

## Figures and Tables

**Figure 1 ijerph-18-12200-f001:**
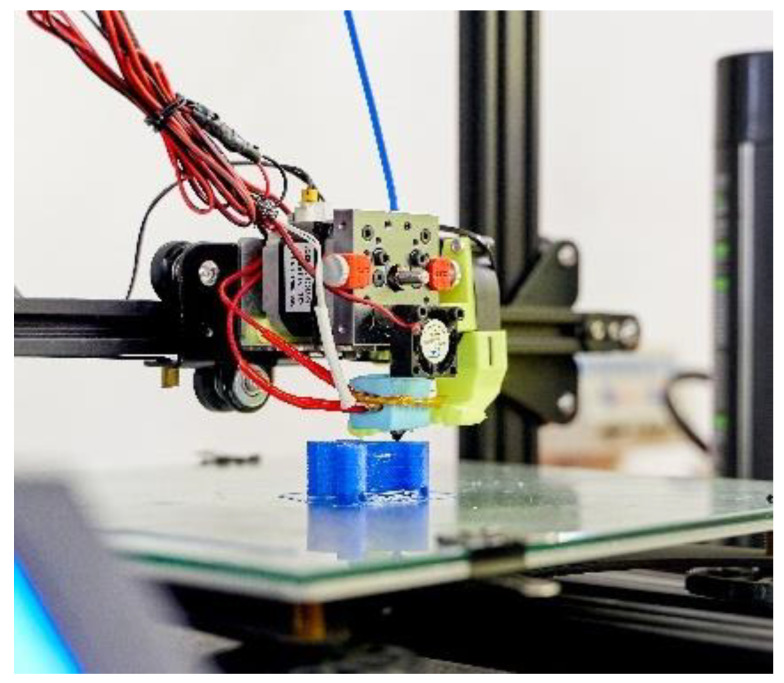
Adapted FDM machine to extrude flexible filaments.

**Figure 2 ijerph-18-12200-f002:**
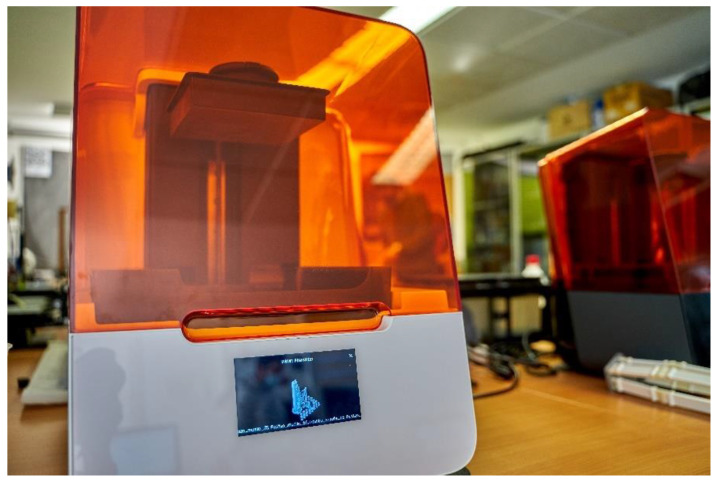
SLA machine ready to print with Tough 2000 Resin.

**Figure 3 ijerph-18-12200-f003:**
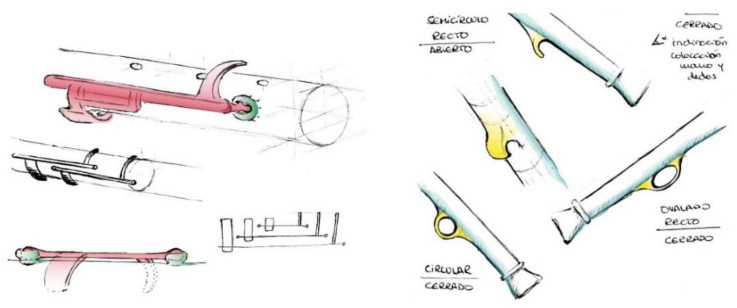
These sketches of Flow show the keys system and grip ring.

**Figure 4 ijerph-18-12200-f004:**
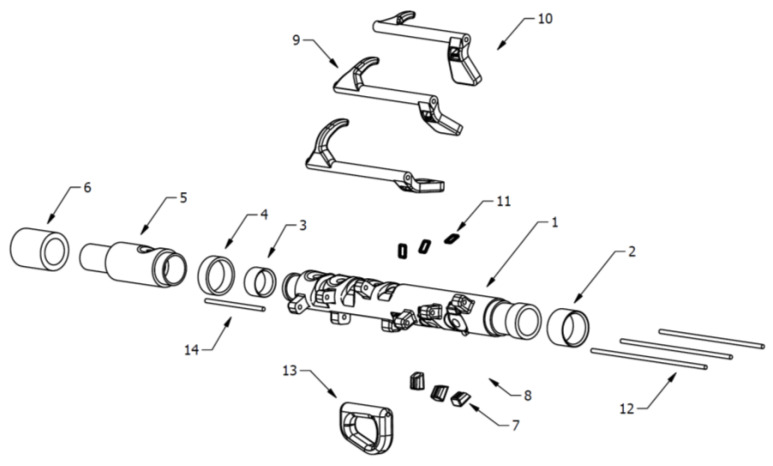
Exploded-view of Flow concept.

**Figure 5 ijerph-18-12200-f005:**
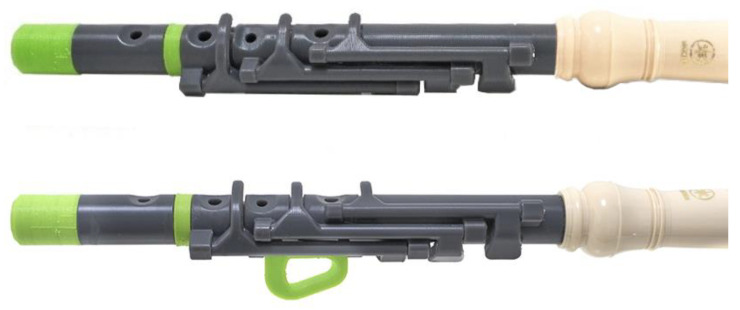
Final design of prototype 2 of Flow.

**Figure 6 ijerph-18-12200-f006:**
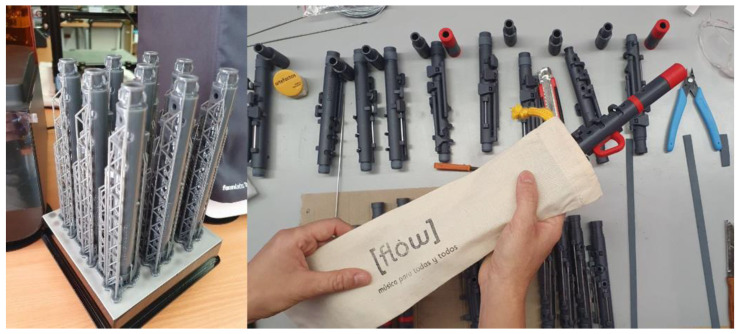
Manufacturing and assembly of Flow.

**Figure 7 ijerph-18-12200-f007:**
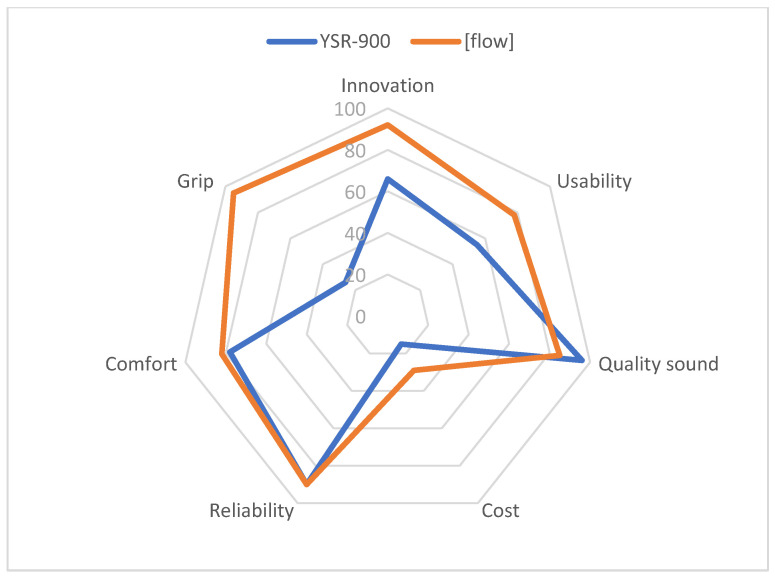
Main properties of the adapted recorders according to musicians’ views.

**Figure 8 ijerph-18-12200-f008:**
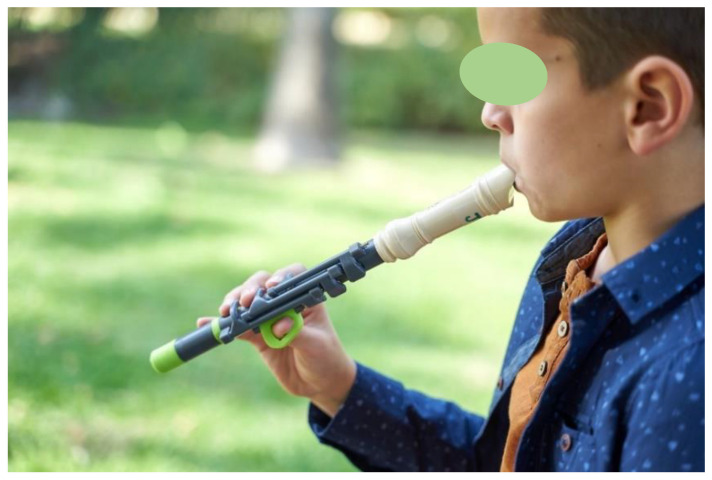
First rehearsals of Jonay (potential user) with Flow.

**Figure 9 ijerph-18-12200-f009:**
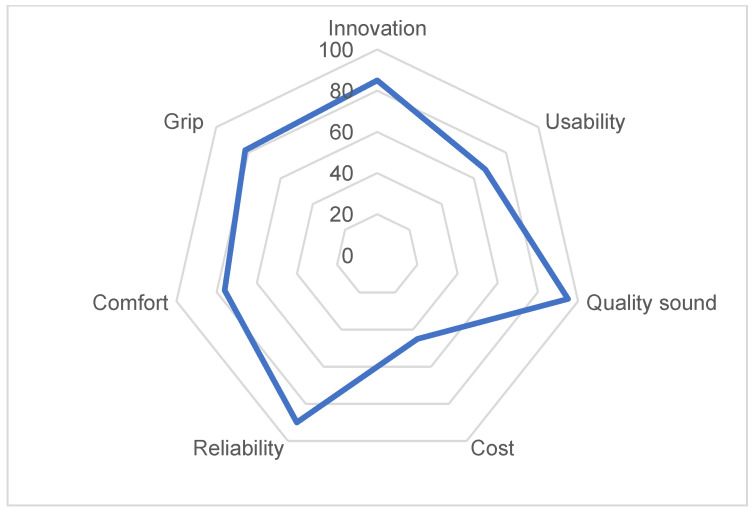
Main features of Flow concept according to users’ assessment.

**Figure 10 ijerph-18-12200-f010:**
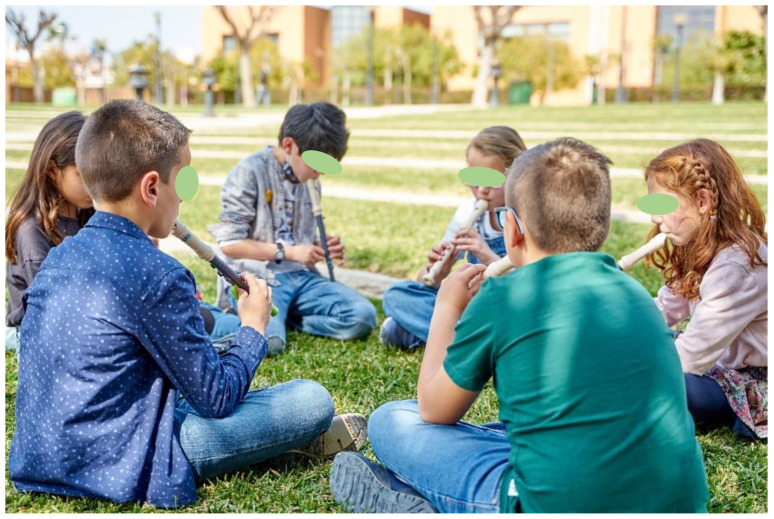
Inclusive school activity using Flow and other recorders.

**Table 1 ijerph-18-12200-t001:** Social profile of the participants.

		Diagnosis							
		Hemiplegia		Agenesis		OBPI		Total	
		No.	%	No.	%	No.	%	No.	%
Genre	Male	2	40	5	62.5	6	85.7	13	65
	Female	3	60	3	37.5	1	14.3	7	35
Type of AT	Recorder Right	3	60	5	62.5	1	14.3	9	45
	Recorder Left	2	40	3	37.5	6	85.7	11	55

**Table 2 ijerph-18-12200-t002:** Comparison between Flow designs proposed and the adapted Yamaha YRS900.

	Yamaha YRS900	Flow Concept
No. of parts (Without mouthpiece)	24 units.	20 units.
Weight (gr.) (Without mouthpiece)	114 gr.	70 gr.
No. of keys	7	3
Manufacturing process	Casting and machining	SLA and FDM
Material	Wood and brass	Tough resin and TPE
Use	All keys can be played using five fingers	All keys can be played with four fingers
Grip	Not adapted. It requires extra support for one-handed people.	Thumb ring and non-slip foot of the body. It can be played with one hand.
Affordances	- Professional quality - Good tuning - Allows students to keep on learning music	- Good tuning - Lightweight - Resistant material - Allows students to keep on learning music
Limitations	- High cost (approx. EUR 400) - Only available in Japan - Difficult to play - Not durable in schools settings	- Complex assembly

**Table 3 ijerph-18-12200-t003:** Notes’ analysis by gStrings app.

		Standard	Yamaha YRS-900		Flow Concept	
Scientific Notation	Musical Notation	Frequency (Hz)	Frequency (Hz)	Deviation (cents)	Frequency (Hz)	Deviation (cents)
C5	C	523.2	524	2.6	538.2	48.9
D5	D	587.3	585.4	−5.6	592.5	15.3
E5	E	659.3	648.7	−28.1	652.7	−17.4
F5	F	698.5	711.3	31.4	693.4	−12.7
G5	G	784	769.8	−31.6	785.0	2.2
A5	A	880	871.1	−17.6	873.9	−12
B5	B	987.8	979.5	−14.6	973.3	−25.6
C6	C′	1046.5	1036.4	−16.8	1036.2	−17.1

**Table 4 ijerph-18-12200-t004:** List of experts in music at schools.

ID	Age	Genre	What Is Your Connection to the Recorder?	How Many Hours Do You Teach in Primary School?	How Many Hours Do You Teach in Secondary School?
1	48	Female	Music teacher in secondary school	1	2
2	44	Female	Music teacher in primary school	2	3
3	49	Female	Music teacher		3
4	38	Female	Teacher	1	
5	43	Male	Teacher	1	1
6	46	Male	Music teacher in secondary school		5
7	57	Male	Music teacher	2	2
8	28	Female	Teacher	1	2
9	28	Female	Teacher	1	2
10	23	Male	Music teacher, Teacher of education, therapeutics, hearing and language	2	1
11	38	Male	Music teacher	5	
12	33	Female	Teacher	1	2
13	42	Male	Music teacher	1	2
14	42	Male	Music teacher in secondary school		2
15	43	Male	Music teacher	1	2

**Table 5 ijerph-18-12200-t005:** Results from participants’ profiles.

User	Type Flow	Diagnostic	Competence	Adaptability	Self-Esteem
1	Left	Hemiparesis	1.25	2.00	0.88
2	Right	Agenesis	1.25	1.00	1.38
3	Left	Agenesis	1.17	1.67	0.75
4	Left	OBPI	0.50	0.83	0.50
5	Left	OBPI	0.58	0.17	0.75
6	Left	OBPI	1.08	0.67	0.88
7	Right	Agenesis	0.83	1.17	0.50
8	Left	OBPI	0.83	1.33	1.00
9	Left	OBPI	0.58	0.17	0.50
10	Left	OBPI	0.33	0.33	0.50
11	Left	Agenesis	0.92	0.83	0.38
12	Right	Agenesis	0.25	0.33	0.38
13	Right	Hemiparesis	0.83	0.50	0.63
14	Left	Hemiparesis	1.00	0.83	1.00
15	Right	Hemiparesis	0.75	1.00	0.50
16	Right	Hemiparesis	1.17	0.50	0.63
17	Left	Agenesis	1.33	0.50	0.50
18	Right	Agenesis	1.17	0.83	0.88
19	Right	Agenesis	1.58	2.00	1.00
20	Right	OBPI	1.58	1.00	1.00
		Mean (SD)	0.95 (0.38)	0.88 (0.54)	0.73 (0.27)

## Data Availability

Data is contained within the article.

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
