# Peer review of "Flow. A Socially Responsible 3D Printed One-Handed Recorder"

_ijerph, 2021, doi:10.3390/ijerph182212200_

Round 1

Reviewer 1 Report

This paper deals with the concept, implementation and evaluation of a one-handed flute, [flow], that is supposed to be affordable and available/accessible/producible by inclusive schools, foundations, institutions, and alike. 

The paper perfectly fits to the special issue topic. The introduction is well-written and contains plausible arguments.

What I am missing is a closer look at previous work of others. Quite some literature has been cited, but the paper does not show to what degree [flow] is cheaper, better accessible, or better in terms of quality.

I find it critical that the paper contains only little research aspects (a questionnaire and a primitive acoustical evaluation) but is mostly promotional, describing a cool innovation using beautiful photos... especially because I am missing the explicit statement that the CAD files for [flow] will be made freely available to encounter the promised "social responsibility".

To be acceptable for publication in a scientific journal, the paper needs to meet two criteria:

1. The acoustical evaluation needs to be improved: You should either carry out playing/listening experiments with experts or carry out objective analyses of the [flow] sound (I added details below).

2. As you promise "social responsibility" in the title of the paper, I need an explicit statement that [flow] is not a commercial product but will be made freely available.

If you can meet these 2 criteria, the paper can become a great contribution to the special issue.

In addition, there are many (mostly minor) things:

line 33: why don't you cite using "[1]" instead of "(World Health Organization, 2019)"?

line 35: "(Instituto Nacional de Estadística, 2008)" is not a proper citation, please add it to the references and use numbers in brackets to cite it

line 47: more accessible than what? You should simply cross out the word "more"

line 58: firstly suggested to you?

lines 86–87: "There are other projects based on the development of one-handed recorders, which entail an adaptation of a common recorder for an accessible price but they show basic quality." -> This seems like your personal opinion, so please indicate that explicitly or refer to a study that has evaluated the quality of one-handed recorders. Plus: Please name examples of one-handed recorders that exist and refer to publications. Plus: The statement seems to contradict your earlier statement that one-handed recorders are only available at high cost by quite renown instrument builders.

line 104: cross out "(OHMI)" 

line 132: SDGs -> please introduce abbreviations before you use them

line 138: PIADS -> please introduce abbreviations before you use them and explain what PIADS is about.

Figure 1: This is a beautiful promotion photo; perfect for a press release. But I consider it inappropriate for a scientific journal as it does not contribute to the understanding of processes or add any other necessary information. The situation might change if you indicate who belongs to an associations, who is a potential user, and who is involved in the technical implementation... something like that. But it is only helpful and appropriate in a scientific journal if it adds information. 

line 154: can you provide literature on human-centered design (some textbook or handbook chapter)? Like [27]

line 160: "The human-centered design consists of three stages:" now this one definitely needs a reference

line 177: The sentence should start with a capital T

lines 183–203: References [30] to [34] are only websites, so I suggest to put the link in a footnote instead of the reference list.

line 216: "0.4mm" needs a space character between value and unit, i.e., "0.4 mm"

Figure 2: Again, the photo looks nice but does not add information. Could you maybe indicate where the flexible filaments extruder is?

line 223: "increase acoustic characteristics" does not make sense. Increase what? Improve certain acoustic characteristics? Which ones? You do provide a reference, but it is impossible to see what it is... a journal paper? Conference paper? Book? Website article?

line 241: What is "ABS"?

Figure 3: Again, not helpful for understanding if you don't highlight e.g.,  where the ultraviolet radiators are, 

Section 2.2.3. Acoustic/musical properties: what was analyzed/examined/compared and how/by whom?

lines 253–254 and line 261: a scale from 3 to +3 ? You probably mean -3 to 3?

line 268: From your citation is appears as if [39] would show the analysis results of surveys that address innovation, usability, quality of sound, reliability, ergonomics, grip and cost of your product... but this is not the case. Why are you referring to [39]?

Table 2: What maintenance is necessary in the Yamaha YRS900?

Figure 4: Though beautiful, I have troubles deriving information from that figure. Please describe it in more detail in the captions plain text and in the captions.

line 342/345, Figure 5: Where do the metal rods and the magnets come from? Do you print them or are they customary and available and cheap everywhere? Many 3D printers cannot print metal, right? Isn't that a major drawback because many institutions cannot print your flute themselves?

Figure 5: Please add some description to the captions

Table 3: I'm pretty sure that a flute can only play the musical note c'' (not C), which equals 523 Hz and C5. More importantly, flutes can be (de-)tuned by moving the head a bit, so why didn't you tune both flutes to one common note first, before analyzing the tuning? Plus: The fundamental frequency is important, but it is more important to know by how many cents the played note deviates from the expected frequency in equal temperament? Have you blown and measured only once? Or do you present a mean value of multiple recordings? Why don't you present f sharp and b flat, too?

line 400: tests (not test)

line 406 and 411: But "usability" and "comfort" only achieved an average rating

line 438: Why all these references? Just listing them at the end seems as if they underlined that the goals of your research was inclusion at schools. This is certainly not the case, so please cite more explicit which reference states what.

line 415–418: "Good acoustics and tuning: the acoustic tests (Table 3) show the high quality of the recorder’s acoustics." This is not true. Even though "good acoustics" are conceptually vague and somewhat subjective, your acoustic study has not demonstrated more than one aspect of tuning (you did not even indicating the detuning in cents).

You should either:
a) let musicians play the recorder and evaluate the quality or
b) identify some objective, acoustic parameters that are related to aspects of perceived sound and then compare a high-quality flute with [flow] and maybe a third, budget flute

Good acoustics consists e.g., of spectral distribution (quantified e.g., by the spectral centroid, spectral spread, and the identification of formant regions), the degree of loudness fluctuations and roughness, how easy and satisfying is the articulation (playing vibrato, tremolo, overblowing, bifurcation, (de)crescendo, ease of excitation, dynamic range, partial covering of holes, and many more), the sound radiation characteristics, ... 

The least thing you should do to estimate acoustical quality is to analyze 

- detuning in cents
- spectral centroid
- spectral spread
- roughness (after Leman, Aures, or Daniel & Weber)

and compare a high-quality flute with [flow] and maybe a third, budget flute.

I suggest to tune [flow] and your reference instrument to the same fundamental note, play all notes and indicate by how much the fundamental frequency deviated from the desired note in cents. Furthermore, I suggest to measure the spectral centroid and the spectral spread.

Patents: It seems counter-productive to patent work that you want to distribute freely to achieve inclusion on a budget level accessible to everyone. Patents indicate commercial interests. Maybe you can explain your motivation to patent your work. And: Is your plan to distribute your CAD-model freely to enable institutions to produce it themselves? Then please explicitly state so. Or is your plan to produce and sell these flutes yourself? If so, please state so. In that case you optimized production costs rather than being "socially responsible" -> every large-scale livestock farming also optimizes production costs without claiming to be "socially responsible".

Furthermore, I only found: ES1275574U / WO2021116514A1 and ES1242275Y

You keep introducing the abbreviations (FDM), (AM) and (SLA) over and over. Please introduce them only once.

Problematic: In Acoustics, AM typically refers to Amplitude Modulation and FDM to Finite Differences Method

As your aim is inclusion for all, I suggest to promote your product over non-academic channels (1. Let your Universities write a press release and let the media jump on that train, 2. Provide the CAD files and a documentation to organizations, hospitals and associations that deal with Amputation, Paralympics, OBPI, Agenesis, Hemiplegia, the inclusion of physically disabled/restricted people etc.)

Author Response

Thank you very much for your comments.

In the attached document I have included all the answers and comments (in red).

Kind regards.

Reviewer 2 Report

The manuscript reports a first attempt at designing, fabricating, and testing a one-handed recorder named [flow] using 3D printing technology. The polymeric prototype of the one-handed recorder is aimed for use by school kids who have difficulties in using both their arms/hands with equal ease. The recorder is designed so that it has less number of parts, is lighter, and user-friendly compared to existing, albeit more professional, one-handed recorders. Moreover, the cost of [flow] is less compared to the available options in the market, albeit it is still high as reported by the users' response in the Psychosocial Impact of Assistive Devices Scale (PIADS) questionnaire. Furthermore, the acoustic features of [flow] are reported to be of good quality.

The work reported here is interesting and has direct societal relevance. Even though it is only the first step, but it seems to be in the right direction. There are some minor comments which needs to be addressed to make this study more clear to the readers. These are explained in the 'Comments' below. After this revision, I recommend that the manuscript could be accepted for publication.

Comments:

  1. Line 20: Please expand PIADS where it is first used, for example in the Abstract.
  2. Line 22: What do the authors mean by 'resistance' here? Is it the airflow resistance or the strength of the material? Please clarify in the revised manuscript.
  3. Line 86: Please specify these 'other projects' in the revised manuscript.
  4. Lines 99-101: This statement is not grammatically correct. Please re-phrase.
  5. Line 132: Please expand SDG where it is first used in the manuscript.
  6. Please check the reference numbers cited throughout the manuscript. Some of them do not seem to be the correct ones. For example, Ref.8 seems to be not directly relevant to this work, Ref. 10 seems to be incomplete, the reference after Ref. 24 is not numbered, etc. Please check the references and their citations in the text thoroughly and correct the mistakes.
  7. Line 177: Please replace 'the assessment' with 'The assessment'.
  8. Lines 216-218: Please expand PLA, TPE, and TPU where they are first used in the manuscript.
  9. Line 241: Please expand ABS where it is first used in the manuscript.
  10. Does the scale in PIADS range from '-3 to +3' or from '3 to +3'? Please check this and clarify in the revised manuscript (for example, in lines 253 and 261).
  11. Line 272: Is OBPI expanded as 'Obstetric Brachial Plexus Palsy' or 'Obstetric Brachial Plexus Injury'? Please check and correct/clarify this in the revised manuscript.
  12. Table 1: 'Recorder Left' seems to be missing. Please correct.
  13. Line 306: Please expand ODS where it is first used in the manuscript.
  14. Line 320: Please replace 'recorded' with 'recorder'.
  15. Table 2: What do the authors mean by 'Maintenance' in the 'Affordances' of the Yamaha recorder? Please clarify in the revision.
  16. Table 2: How was the durability of [flow] tested? Please explain in the revision.
  17. Fig. 5: Why is the number of parts of [flow], mentioned as 18 in Table 2, different here? Please clarify.
  18. Line 360: It is not clear if SLA was used to make the adapted recorder, since it is mentioned that '...can be manufactured by SLA...' -- please clarify.
  19. Fig. 6: It would be helpful to show one additional view of the prototype where the grip ring is also seen.
  20. Table 3: YRS or YSR? -- please check and correct.
  21. Table 3: Were there repetitive tests to ensure that the reported values are repeatable?
  22. Fig. 8:  It is understandable that a user could perhaps more easily assess cost, grip, and comfort, whereas 'reliability' and 'innovation' appear as vague parameters on a first glance. It wold be helpful if the questionnaire is also included, perhaps in the Appendix. For example, this would be helpful to clarify how parameters like 'reliability' are assessed by the users.  
  23. Line 396: How are the sub-scale scores in Table 4 interpreted as high scores?
  24. Line 414: Please replace 'is being design' with 'is designed'.
  25. Line 455: Please cite the relevant studies here.
  26. Author contributions: The initials of the authors seem to not match with the authors list in the first page. Please check and correct.
  27. Line 502: Why is the date of approval not included?

Author Response

(The authors gave the same response as above.)

Round 2

Reviewer 1 Report

Dear authors, thank you for responding to my comments and revising the manuscript accordingly.

This is a fine paper and it has improved through your revision. I still have one major concern about the "Acoustic results" and I see some minor issued:

Sections 3.1.3 and 3.2: Please indicate in the plain text that 15 or 20, respectively, participants completed the questionnaire.

Line 434: "an" should read "and"

Sect. 3.1.2. Acoustic results: This analysis is neither acoustically, nor musically sound. I understand that it is not helpful to tell you what you should have done (namely: tuning all instruments so that the fundamental frequency of the lowest note is exactly right), even though the experiment could be repeated within one hour if you have the flutes at hand. However, I see that acoustical analysis is not that important for the purpose of your paper. But the least thing you should do is calculate the deviation between the target frequency (you called it "standard") and the measured frequency ("Yamaha YRS 900" and "[flow] concept") in cents. I could do it manually, or you do it, e.g., by copy-pasting the frequencies into that online converter ("Cent value-determination of an interval"):

http://www.sengpielaudio.com/calculator-centsratio.htm

and add these deviations to table 3.

And again: For Yamaha you mostly indicate a range, for [flow] you indicate a single frequency, so please briefly explain how come: Have you recorded the Yamaha multiple times? Or does it offer multiple ways to produce the same note?

concerning the "social responsibility": From what you wrote to me it seems that [flow] is indeed not a commercial product, as you produce it without extra costs ("only material and technical staff to fabrication"). Please just make that explicit in the paper. And by "freely available" I meant the CAD file, so that peole could print it their own. I'm still wondering what the patents are for. Are you planing to sell licenses?

Anyways, you should add the frequency deviation in cents to the table, that's all, but that's enough to call it "major", because I want to make sure that you made it right so that the acoustical analysis has achieved the minimum required soundness.

Best regards.

Author Response

Again, thank you very much for your comments.

In the attached document I have included all the answers and comments (in red).

Kind regards.

Round 3

Reviewer 1 Report

Thank you a lot.